# Identifying Biomarkers for Remyelination and Recovery in Multiple Sclerosis: A Measure of Progress

**DOI:** 10.3390/biomedicines13020357

**Published:** 2025-02-04

**Authors:** Vito A. G. Ricigliano, Silvia Marenna, Serena Borrelli, Valentina Camera, Edgar Carnero Contentti, Natalia Szejko, Christos Bakirtzis, Sanja Gluscevic, Sara Samadzadeh, Hans-Peter Hartung, Krzysztof Selmaj, Bruno Stankoff, Giancarlo Comi

**Affiliations:** 1Neurology Department, Pitié-Salpêtrière Hospital, APHP, 75013 Paris, France; ricivito@hotmail.it (V.A.G.R.); bruno.stankoff@aphp.fr (B.S.); 2Neurology Unit, GHNE—Paris Saclay Hospital, 91400 Orsay, France; 3Experimental Neurophysiology Unit, Institute of Experimental Neurology (INSPE), IRCCS-San Raffaele Scientific Institute, Università Vita-Salute San Raffaele, 20132 Milan, Italy; marenna.silvia@hsr.it; 4Neuroinflammation Imaging Laboratory (NIL), Institute of NeuroScience, Université Catholique de Louvain, 1348 Brussels, Belgium; serena.borrelli.sb@gmail.com; 5Department of Neurology, Hôpital Erasme, Hôpital Universitaire de Bruxelles, Université Libre de Bruxelles, 1070 Brussels, Belgium; 6Nuffield Departement of Clinical Neuroscience, University of Oxford, Oxford OX1 2JD, UK; valentina.camera@gmail.com; 7Department of Neuroscience, Biomedicine and Movement Science, University of Verona, 37129 Verona, Italy; 8Neuroimmunology Unit, Department of Neurosciences, Hospital Alemán, Buenos Aires C1425FQB, Argentina; junior.carnero@hotmail.com; 9Department of Clinical Neurosciences, University of Calgary, Calgary, AB T2N 1N4, Canada; natalia.szejko@gmail.com; 10Department of Bioethics, Medical University of Warsaw, 02-091 Warsaw, Poland; 11Multiple Sclerosis Center, Second Department of Neurology, Aristotle University of Thessaloniki, 541 24 Thessaloniki, Greece; bakirtzischristos@yahoo.gr; 12Neurology Clinic, Clinical Centre of Montenegro, 81000 Podgorica, Montenegro; sanja.v.gluscevic@gmail.com; 13Experimental and Clinical Research Center, Charité—Universitätsmedizin Berlin, Corporate Member of Freie Universität Berlin and Humboldt-Universität zu Berlin, 12203 Berlin, Germany; 14Institute of Regional Health Research and, Institute of Molecular Medicine, University of Southern Denmark, 5230 Odense, Denmark; 15The Center for Neurological Research, Department of Neurology, Næstved-Slagelse-Ringsted Hospitals, 4200 Slagelse, Denmark; 16Brain and Mind Center, University of Sydney, Sydney 2050, Australia; hartunghp@gmail.com; 17Department of Neurology, Palacky University Olomouc, 779 00 Olomouc, Czech Republic; 18Department of Neurology, Medical Faculty, Heinrich-Heine University, 40225 Düsseldorf, Germany; 19Department of Neurology, University of Warmia & Mazury, 10-719 Olsztyn, Poland; kselmaj@gmail.com; 20Paris Brain Institute, ICM, CNRS, Inserm, Sorbonne Université, 75005 Paris, France; 21Department of Neurorehabilitation Sciences, Casa di Cura Igea, 20129 Milan, Italy; comi.giancarlo@unisr.it

**Keywords:** multiple sclerosis, remyelination, recovery, biomarkers

## Abstract

**Background:** Multiple sclerosis (MS) pathology is characterized by acute and chronic inflammation, demyelination, axonal injury, and neurodegeneration. After decades of research into MS-related degeneration, recent efforts have shifted toward recovery and the prevention of further damage. A key area of focus is the remyelination process, where researchers are studying the effects of pharmacotherapy on myelin repair mechanisms. Multiple compounds are being tested for their potential to foster remyelination in different clinical settings through the application of less or more complex techniques to assess their efficacy. **Objective:** To review current methods and biomarkers to track myelin regeneration and recovery over time in people with MS (PwMS), with potential implications for promyelinating drug testing. **Methods**: Narrative review, based on a selection of PubMed articles discussing techniques to measure in vivo myelin repair and functional recovery in PwMS. **Results**: Non-invasive tools, such as structural Magnetic Resonance Imaging (MRI) and Positron Emission Tomography (PET), are being implemented to track myelin repair, while other techniques like evoked potentials, functional MRI, and digital markers allow the assessment of functional recovery. These methods, alone or in combination, have been employed to obtain precise biomarkers of remyelination and recovery in various clinical trials on MS. **Conclusions**: Combining different techniques to identify myelin restoration in MS could yield novel biomarkers, enhancing the accuracy of clinical trial outcomes for remyelinating therapies in PwMS.

## 1. Introduction

In recent decades, significant progress has been made in understanding the biology of physiological myelination during the developmental phases of the central nervous system (CNS) in humans, particularly in the role of oligodendrocytes and their interactions with other CNS cells [1,2]. Similar processes occur in disease conditions, such as multiple sclerosis (MS), where remyelination follows a demyelinating insult. Remyelination shares some features with developmental myelination and progresses through multiple stages [3]. These stages are marked by morphological changes and the expression of specific myelin markers, starting with the activation of oligodendrocyte precursor cells (OPCs), which proliferate, migrate, and differentiate into mature cells [3,4].

Various biological pathways, especially those involving CNS inflammation [5], can inhibit or promote cell differentiation and remyelination, influencing the prevention of axonal degeneration and the restoration of normal function [6]. Chronic inflammation in focal lesions, largely driven by microglial activity, is associated with reduced remyelination potential. Studies show that myelin repair is less common in mixed active–inactive lesions compared to other types [7]. Chronic lesions also experience repeated demyelination, particularly in previously remyelinated areas, which decreases the lesion’s capacity for repair over time [8,9]. Conversely, microglia in areas of new myelin deposition can promote regeneration [10]. Neuronal electrical activity and microglia–axon interactions at the nodes of Ranvier contribute to efficient remyelination [11]. These opposed effects of microglia highlight the complexity and heterogeneity of their responses, which depend on factors such as location, metabolism, and the state of ongoing demyelination/remyelination processes. This moves away from a simplistic pro- and anti-inflammatory view of microglia, as seen in animal models, toward a broader understanding of reparative and toxic phenotypes in the human brain [10,12]. Furthermore, among other mechanisms influencing proper brain myelination is the efficient clearance of waste products through meningeal lymphatics, which promotes oligodendrocyte survival, as recently suggested [13].

In the study of human remyelination, transitioning from biological insights to clinical application often involves analyzing post mortem data. Remyelination is typically seen in “shadow plaques”, areas characterized by less intense myelin staining due to a thinner myelin sheath. In MS, remyelination is heterogeneous across brain regions and limited to a minority of patients [14,15]. While this variability in remyelination capacity is well-documented in the brains of people with MS (PwMS) [15], it is less explored in the spinal cord. Conventional histopathological methods, such as myelin pallor (luxol fast blue) or myelin thinning, may fail to capture the complete dynamics of repair, potentially overestimating or overlooking certain remyelinated areas—especially when remyelination is nearly complete and resembles normal myelin [16]. Such a condition can be found in sites where significant migration of neural precursor cells from the adult subventricular zone induces extensive remyelination of lesional tissue [17]. Furthermore, and more importantly, post-mortem techniques do not allow the acquisition of temporal information on myelin repair over time. Therefore, it is essential to have in vivo strategies to better detect, quantify, and monitor the spontaneous capacity of remyelination in PwMS. This knowledge would facilitate the development of targeted strategies to enhance the process through external interventions.

In recent years, a variety of pharmacological compounds have been identified as potential tools to foster remyelination in MS, with efficacy demonstrated in both in vitro and in vivo models [6,18]. However, urgent unmet needs remain in understanding the consequences of spontaneous and induced remyelination in terms of axonal protection, its potential in different forms of the disease, and its ability to limit clinical progression. Disentangling the mechanisms that enable remyelination in certain subjects and brain regions, but not in others, would facilitate the development of targeted strategies to enhance the process through external interventions [19].

Studies suggest that myelin sheath reconstitution requires preserved axons to occur properly and is more effective in early MS, with efficiency decreasing with age [20]. Although timely and accurate remyelination is known to restore function [21,22], the specific biological contribution of remyelination to disability recovery, particularly after relapses, is poorly understood [23]. These unsolved aspects critically impact the development of studies testing the efficacy of potential promyelinating compounds in MS. Regarding clinical trials, major questions arise, concerning, among others, patient selection, study methodology, tool standardization, and outcomes.

In this review, we specifically focus on in vivo techniques that can be applied in PwMS to identify and monitor remyelination and repair over time, as well as quantify its impact on halting neurodegeneration and promoting functional recovery. Structural biomarkers, based on magnetic resonance imaging (MRI) and positron emission tomography (PET) with compounds that stain myelin, offer the ability to dynamically track myelin loss and repair. Functional metrics, such as functional MRI (fMRI) and evoked potentials (EPs), enable the functional assessment of remyelination and its capacity to restore neuronal function. Fluid-derived biomarkers allow us to follow up on structural recovery over time, with potential implications for treatment decisions. A combination of these biomarkers could eventually be used as a composite outcome in remyelinating trials (Table 1 and Figure 1).

## 2. Techniques to Study Recovery

### 2.1. Magnetic Resonance Imaging (MRI)

The most widely applied techniques for quantifying myelin content in the CNS in vivo are based on MRI [24], through sequences like the myelin water fraction (MWF) [25], diffusion-weighted imaging (DWI) [26,27], and magnetization transfer ratio (MTR) imaging [28].

The MWF model for studying myelin is based on the principle of different relaxation times for the water trapped within myelin layers (myelin water) and the water in other compartments, such as the intracellular and extracellular spaces [29]. In the multicomponent T2 neuroimaging of MWF, different components are revealed at different T2 times: short T2 corresponds to the myelin–water contrast, intermediate T2 reflects intra- and extracellular water, and long T2 corresponds to the signal from cerebrospinal fluid (CSF) [29]. By measuring the fraction of the total water signal attributed to myelin water, this technique provides a specific marker of myelin integrity. MWF has several disadvantages that limit its clinical use since it involves intricate multi-echo T2 relaxation sequences, which require precise timing and calibration. This complexity extends to the analysis phase, where advanced algorithms must differentiate between myelin water and other water compartments [30].

DWI is also a valuable MRI technique for studying myelin content and WM integrity in the brain. This method provides structural information on the body’s tissue microstructure based on the different motion properties of water molecules in extracellular versus intracellular compartments. Among diffusion-weighted techniques is diffusion tensor imaging (DTI), which relies on the principle of anisotropy of water diffusion in brain white matter (WM) since aligned fibers “restrict” free molecular movement. Therefore, DTI can provide micro-architectural details of WM tracts and indirect insights into myelin integrity and distribution [27]. The closest proxy of myelin integrity derived from DTI is radial diffusivity (RD), which describes water movement perpendicular to axonal tracts, possibly reflecting (although not univocally) [31]) prevalent microstructural changes resulting from myelin loss [32]. However, even though DTI gives crucial information about WM integrity, it does not directly measure myelin content [33]. Instead, it infers myelin integrity from water diffusion patterns, which can also be affected by factors such as axonal damage and inflammation [33,34,35]. Neurite orientation dispersion and density imaging (NODDI) is a more recent diffusion imaging method, based on a multi-compartment model, from which the neurite density and the orientation dispersion index can be extracted [36]. These indices provide more granular information on the microstructural origin of diffusion anisotropy, with the former having a good correlation with myelin stain [37].

MTR is another myelin MRI metric that measures the exchange of magnetization between free water protons and protons bound to macromolecules, such as myelin. This interaction affects the MRI signal and can be quantified to provide insights into tissue composition. MTR is advantageous because it is relatively straightforward to analyze and faster compared to other myelin imaging methods. However, it is influenced by various factors beyond myelin, such as inflammation and tissue hydration/edema, thus complicating the interpretation of results [34,35,38].

In addition to these techniques, quantitative susceptibility mapping (QSM) and magnetic resonance spectroscopy (MRS) offer advanced methods for evaluating complex tissue changes in MS. QSM provides an assessment of tissue magnetic susceptibility, which is influenced by various factors including myelin content and iron deposition. QSM is particularly valuable in the context of MS as it can help differentiate between demyelination and other pathological processes, such as the iron accumulation present in chronically activated microglia. By mapping these susceptibility changes, QSM offers a more nuanced understanding of tissue modifications, making it a powerful tool for both clinical and research applications [39,40]. With regard to remyelination, neuropathologic imaging data suggest that QSM accurately identifies fully remyelinated areas as hypo-/iso-intense [41]. Regarding susceptibility-sensitive sequences more broadly, recently, a biophysical model applied to multi-echo T2*-data and T2-data based on χ-separation was proposed to assess myelin in WM and cortical lesions, distinguishing its contribution from that of iron to the susceptibility signal [35].

### 2.2. Positron Emission Tomography (PET)

PET with radiolabeled compounds directed against specific cellular and molecular targets allows in vivo tracking of different mechanisms of damage and recovery in MS, including myelin loss and repair, chronic inflammation, neuronal injury, and neuroplasticity [42].

Regarding myelin PET, the most widely used tracers are amyloid-binding compounds. Indeed, following the first observations that 1,4-bis(p-aminostyryl)-2-methoxy benzene (BMB), a known amyloid-binding agent, was able to stain myelin in experimental models of MS [43], many other amyloid markers showed the same capacity, among which is the Pittsburgh compound B (PiB). This radiotracer was used in the first study exploring myelin dynamics with PET in PwMS [44], providing information on the intra- and inter-subject heterogeneity of remyelination in MS.

In the research on myelin PET, an interesting field is the development and improvement of new tracers to increase the quality of acquired images. An investigation comparing different PET radiolabeled compounds for myelin, performed in baboons, found that the best contrast between gray matter (GM) and WM is obtained with Florbetapir [45]. Therefore, future PET studies in humans should preferentially use this radiotracer to obtain better images, as confirmed by its first applications in PwMS [46]. Using these higher-quality tracers, phase II clinical studies on remyelinating treatments in MS could benefit PET outcomes to define myelin dynamics over time [6]. Beyond amyloid tracers, other compounds are being successfully applied to study myelin dynamics in vivo, in particular those directed against the axonal potassium channel, which becomes exposed upon demyelination. The binding to this channel would therefore give a positive signal, bearing double information: where demyelination is located and whether the axons underneath are still there. Encouraging data applying these compounds are accumulating in animal models, healthy controls [47,48,49], and PwMS. Some other PET tracer candidates targeting oligodendrocyte metabolic dysfunction even in the absence of demyelination, e.g., GPR17 and MCT1, are also being developed [50].

Moving to neurodegeneration and protection, PET can also be used to identify neuronal loss, e.g., by binding the GABA-A receptor that is ubiquitously expressed in neurons and may be decreased in the case of early neuronal damage in PwMS [51]. More promising radiotracers, i.e., directed against the synaptic vesicle protein 2A [52], could be able to detect an early reduction in synaptic density [52,53].

The limitations of PET application in exploring MS pathology are, among others, its cost, the need for radiotracers, and the limited access to this technique across the world. To overcome this issue, reconstructing PET imaging through MRI could be a solution [54]. In this direction, deep learning approaches combining simple MRI sequences, such as T1 and FLAIR, were recently shown to reproduce quite accurate myelin and PET images.

Altogether, these investigations demonstrate that, using different radiotracers, PET is a powerful and specific tool to identify in vivo the processes of tissue damage and repair in MS, and, more importantly, to stratify subjects based on different pathophysiologic mechanisms. This offers the unique possibility to extract patient-specific profiles of damage and recovery that could eventually be used as outcome measures in phase II clinical trials of promyelinating and neuroprotective compounds.

### 2.3. Functional Metrics (fMRI, EPs, OCT, and Digital Markers)

Functional MRI (fMRI) is a technique that measures the activation of different brain regions by tracking changes associated with tissue blood flow in relation to neuronal activity [55]. Based on this principle, fMRI allows us to extract information on functional connectivity networks across different zones and their reorganization in disease conditions. Relevant information on adaptive network modifications can be obtained through task-based or resting-state fMRI. In the context of MS, fMRI modifications reflecting changes in neural networks have been described following optic neuritis and may be part of recovery mechanisms occurring in the affected brain [56,57]. Studies have shown that early disease phases are characterized by compensatory increases in fMRI activity, followed by gradual exhaustion at later stages [58]. Regarding fMRI-derived markers of repair, some works have shown good correspondence between metrics such as functional connectivity and intracortical myelin content [59].

Among the other non-invasive measures used to functionally track damage and repair in MS, EPs can identify demyelination early before symptom appearance thanks to their sensitivity in detecting dysfunction. They provide information in advance that a lesion can become symptomatic in the future. Indeed, electrophysiological tools can reflect the excitability and functionality of the pathway under investigation. For example, by focusing on the visual system it is possible to investigate its features by non-invasive VEPs. VEP latency reflects the velocity of the signal conduction along the visual pathway [60]. Indeed, a pre-clinical study on the experimental autoimmune encephalomyelitis (EAE) model demonstrated the usefulness of electrophysiological recordings to detect the presence or absence of myelin in the optic nerve [61].

Considering these aspects, VEPs can be also applied to detect remyelination by assessing the recovery of VEP latency. In some cases, the recovery or prevention of VEP delay passes through the restoration of a normal amount of myelin. For instance, by applying VEP recording in the toxic de-/remyelination cuprizone model, researchers demonstrated demyelination and remyelination in the optic nerve using VEP delay; complete recovery was then validated by histology [62]. Additionally, VEPs may be used as biomarkers to evaluate remyelination after treatment. In a study in which transcranial direct current stimulation (tDCS) was applied to the cuprizone model, for example, VEP latency reflected faster myelin recovery in treated mice than in the control.

One of the interesting features of this neurophysiological tool is the ability to show myelin function and not just the amount of myelin in the visual pathway. A pre-clinical experiment in the EAE model with tDCS demonstrated that applying inhibitory neuromodulation reduced inflammatory activity even before the occurrence of the VEP delay, preventing the future appearance of latency delay and myelin dysfunction. In these experiments, myelin structure was investigated in detail. Researchers found that the treatment effect did not seem to be correlated to the rough amount of myelin seen by Luxol fast blue, but rather to myelin functional integrity [63]. Indeed, stimulated mice showed lower disruption of paranodes (more complete paranode domains and fewer single paranodes) and, globally, more adherent myelin. Therefore, VEPs can be used as a biomarker for myelin de-/remyelination thanks to their sensitivity in detecting and measuring myelin function and not just the quantity of myelin in the tissue.

To gain a complete overview of the disease progression in terms of both damage and repair, VEPs can be combined with other non-invasive electrophysiological recordings such as motor EP (MEP) or somatosensory EP (SSEP). In RRMS, MEP may predict, by 2 years, the appearance of motor symptoms along the motor path explored, with 80% accuracy [64]. In terms of measuring progression, a more recent study found that monitoring PwMS over time through EP was able to detect a «silent» progressive increase in the latency of VEP, SSEP, and MEP [65]. To note, this constant increase reaches a plateau effect when axons are so degenerated that an electrophysiological response can no longer be induced, particularly in the long axons of lower limbs [66].

Apart from predicting future symptoms and silent progression, in general, EPs can provide information on the presence of preserved axons through the spared signal of Eps, reflecting the fact that axons are still there [67]. Indeed, the amplitude of EPs is associated with axonal degeneration. This may be a neurophysiological way to indirectly measure the axonal reserve without the need for imaging. Evidence was presented in pre-clinical and clinical studies. In pre-clinical settings, VEP recording in healthy mice demonstrated good sensitivity in detecting amplitude change in the visual pathway after non-invasive brain stimulation. In clinical investigations, combining MEP and SSEP with VEP in a composite score allows the identification of subjects who are going to respond and get better through rehabilitation, corresponding to those with lower abnormalities in EPs [67]. Of note, MEP monitoring has a limited value in the lower limbs in progressive MS, as they could be absent in 75% of subjects [68]. In such cases, shorter axonal paths like those of upper limbs, together with walking speed, could be the best option to use as a surrogate method because it is more viable and suitable for long-term monitoring.

The use of a sensitive biomarker for detecting early damage could allow immediate treatment, thus reducing demyelination and axonal degeneration. Unfortunately, as mentioned before, VEP amplitude is only partially associated with axonal loss. Indeed, there may be different mechanisms inducing this type of damage that VEPs are not able to discriminate. The amplitude reduction could reflect conduction blocks or partial axonal loss, without defining the damage location. In this scenario, neurofilament quantification can be a sensitive marker to monitor axonal degeneration [69].

For these and other reasons, over the years, a non-invasive imaging tool to explore the visual system was implemented, which is able to measure the presence of axons in the retina: Optical Coherence Tomography (OCT). OCT allows the measurement of the thickness, presence, and degeneration of retinal axons. Preserving retinal layers reflects neuro-axonal integrity, thus behaving as a biomarker of neuronal survival and myelin repair at the same time, as shown in some clinical trials [70,71,72]. Through scanning of the eye and automatic analysis of the images, it is possible to measure the retinal nerve fiber layer (RNFL) and the ganglion cell layer (GCL). The blending of VEPs and OCT permits the evaluation of the damage and/or recovery of the visual pathway with more specificity. In disease status, it could occur that VEP and vision improve while RNFL and GCL worsen due to neuroaxonal degeneration and loss in the retina [73]. Moreover, OCT allows us to distinguish more pronounced loss in RNFL in PwMS with evidence of disease activity, compared to those with no evidence of disease activity, NEDA-3 [74]. When assessing improvement, the inner retinal layer is the one that electively recovers after steroid treatment or the introduction of immunomodulators [75,76]. The possibility of integrating functional and structural biomarkers to investigate the visual system may therefore give a more sensitive description of the damage in MS and of treatment efficacy.

Lastly, a growing interest has been shown in recent years in so-called «digital markers» to remotely monitor chronic diseases like MS. These metrics have the advantage of allowing a more granular assessment of clinical status compared to 6-month or yearly visits. Moreover, they reflect the scales performed by raters in clinical settings with good accuracy (e.g., 9HPT, T25FW, or oral SDMT) [77]. Additionally, they offer the possibility of overcoming two typical problems of active testing. First, the tendency to perform better when tested in a lab compared to when subjects are at home and act according to their habits [78], and second, the effect of getting tired and abandoning active testing after one month on average [79]. In PwMS, a good association has been shown between the continuous assessment of step counts through wearable sensors and clinical progression [80]. Furthermore, sensors can also detect more subtle markers of walking ability, such as peak cadence and other qualitative features of steps, which are well-correlated with disability [81].

This short overview shows that different functional, structural, and digital metrics offer complementary information on clinical worsening and improvement in MS, therefore combining them would be the best solution to accurately track damage and recovery in PwMS.

## 3. Outcome Biomarkers for Therapeutic Interventions to Enhance Recovery

### 3.1. Vision

Two aspects are relevant regarding regeneration of the visual system: axonal regrowth and remyelination. Regarding axonal regrowth, stimulation methods such as non-invasive electrical stimulation (ES1) [82] are being evaluated for their potential to improve visual impairment. In the last 10 years, ES has been applied in preclinical and clinical research thanks to the possibility of avoiding constraints imposed by drugs, such as a lack of selectivity to specific lesions, allergic reactions, or drug resistance development [83]. Interestingly, non-invasive ES can influence neuronal polarization as well as oligodendrocyte activity [84], improving their differentiation. Moreover, it has been shown that increased oligodendrocyte extracellular vesicle release ameliorates axonal survival [85]. Unfortunately, ES molecular mechanisms are only partially known and need further investigation: preclinical studies are currently being performed to discover and characterize mechanisms for axonal degeneration prevention and remyelination. Therefore, drug-based interventions currently seem to remain the quickest way to enhance recovery in the visual system. To evaluate which drug has major effects on the visual system, improving remyelination and/or axonal regeneration, two main pathways are being considered: optic nerve functionality and eye movements. Concerning the first, distinct visual outcome measures for ON in clinical trials have been evaluated, with VEP latency being the most successful metric [71]. Beyond pre-clinical studies, VEP recordings have already been applied in clinical setting. Indeed, this electrophysiological tool is relatively simple, inexpensive, and immediately available in clinical investigation. These features allow the use of VEPs to follow up patients. As an example, VEP recording in follow-up was able to detect spontaneous recovery 1–3 months [86] after optic neuritis (ON) [87] by evaluating the state of surviving, functioning myelin [88]. VEP improvement has already been used as an outcome measure in two clinical trials on MS, with Opicinumab for acute ON, and Clemastine for chronic ON [70,71]. Apart from randomized studies, even in a real-world setting, monitoring VEP recovery in PwMS can inform us of the spontaneous ability to remyelinate following ON thanks to the innate mechanisms of myelin repair, which can be potentiated by classical immunomodulatory molecules. After Natalizumab, for instance, most subjects remain stable or silently improve their VEP delay [89], and few show silent progression. Similar data identified the silent improvement in the latency of VEP under Teriflunomide [90] and in multifocal VEP under Alemtuzumab [91].

Unfortunately, some limitations should be considered in evaluating the functioning in clinical trials and VEP application. First, there is bias related to optic nerve pathway involvement since it is not possible to directly translate what happens in the optic nerve into other parts of the brain in PwMS. Moreover, non-responders (axonotmesis) and the inability to predict responders are other limitations. For these reasons, different models were developed with the aim of skipping the bias of the optic nerve pathway and predicting treatment responders.

The saccadic eye movement model consists of a pair of eye movements toward two stimuli presented in quick succession [92]. This model could be used as a novel outcome measure for remyelination trials in MS as it can extrapolate information going from demyelination of the medial longitudinal fasciculus in the brainstem to quantitative assessment of cortical networks controlling saccadic eye movements in MS. Internuclear ophtalmoparesis (INO) metrics, with abduction nystagmus evaluating horizontal eye position, are a relevant and highly reproducible measure, especially considering that INO is relatively prevalent in PwMS [93,94]. In this context, horizontal saccadic movement was recorded in a crossover trial with fampridine to demonstrate the possibility of predicting treatment response. The results of a randomized, double-blind, placebo-controlled, crossover trial in PwMS and INO showed that fampridine improves saccadic eye movements linked to INO in PwMS [95]. Thus, the treatment response to fampridine may gauge patient selection for inclusion in remyelinating strategies in MS, using saccadic eye movements as the primary outcome measure. Currently, the MS center in Amsterdam is recruiting PwMS to evaluate remyelination in the RESTORE trial (NCT05338450) with Clemastine fumarate 8 mg daily versus placebo for six months. The primary endpoint will be the variation of the area under the curve of the Versional Dysconjugacy Index (VDI) measured through infrared oculography.

In conclusion, evaluating remyelination in the visual system by functional measures is a helpful way to investigate treatment efficacy. To obtain more successful trials, it is important to investigate both the afferent (pVEP) and efferent (VDI area under the curve) visual pathways to obtain confirmation of the results.

### 3.2. Brain

In clinical trials aiming at assessing the effects of currently approved disease-modifying therapies or putative promyelinating compounds on brain remyelination, MRI was the most widely applied tool. In the Phase 3 DEFINE study [39], the authors measured changes in MTR as a potential indicator of myelin density in the brain tissue of patients with relapsing–remitting MS (RRMS) treated with dimethyl fumarate (DMF). In this analysis, MTR increases in brain tissue likely reflected higher myelin density in response to DMF. Similarly, the more recent EXPAND trial [96] investigated the effect of siponimod on brain atrophy, but also on MTR changes. Compared with placebo, siponimod significantly reduced brain atrophy progression over 12 and 24 months of treatment and was associated with improvements in brain tissue integrity and myelin density. Additionally, the ReBUILD trial, a double-blind, randomized, placebo-controlled remyelination study, showed a significant reduction in visual evoked potential (VEP) latency in PwMS treated with clemastine [97]. The study documented an increase in MWF in the normal-appearing WM of the corpus callosum, providing direct, imaging-based evidence of drug-induced myelin repair. It also highlighted that significant myelin repair occurs outside of MS lesions, suggesting that myelin assessments should also focus on areas outside focal lesions. A clinical trial assessing the effects of the retinoid-X receptor agonist Bexarotene in promoting brain remyelination, based on positive results obtained in mice by targeting this pathway, was conducted in recent years, setting changes in myelin-sensitive MRI metrics and electrophysiology as outcomes. Despite a negative primary efficacy outcome, with no significant variations in MTR values within demyelinated lesions, a global MTR improvement was observed in the MS brain, reflecting greater remyelination. Interestingly, a significant positive and age-dependent MTR increase was found in gray matter lesions. Additionally, Bexarotene succeeded in reducing VEP latency in patients with ON [72].

Spatially, myelin recovery in the MS brain can be explored in different locations, going from focal WM and GM lesions to normal-appearing tissues. These dynamic evaluations can benefit from both MRI and PET measurements.

Regarding GM remyelination, some works have linked dynamic changes in cortical myelin to longitudinal disability, emphasizing its contribution to clinical progression [98,99]. MTR imaging has been recently applied in studying the individual potential of spontaneous myelin repair at this level. A longitudinal multicenter MTR study enabled voxel-level classification of cortical myelin content changes in PwMS, revealing heterogeneous remyelination profiles across individuals, and a weak correlation between WM and GM remyelination processes, suggesting separate mechanisms in these two different areas [98,99]. Interestingly, this research also showed that myelin repair allowed greater cortical volume preservation (seen as reduced GM atrophy in remyelinated areas), but the association was only present in the first five years of the disease. This result encourages the early application of promyelinating interventions during the MS course to produce a neuroprotective effect, possibly related to the higher number of viable and still recoverable neurons within the affected tissue. Additionally, looking at remyelination potential at different cortical depths, it was shown that the superficial cortical layer, close to meningeal CSF, has lower myelin repair compared to deeper layers. This defines a gradient of remyelination failure in relation to the proximity of CSF in the subarachnoid space [100].

In the WM, other than with MRI as performed in the trials described above and in multiple research studies [26], remyelination was also explored with PET using myelin-binding tracers. A pioneer research study assessed the dynamics of myelin repair in PwMS with the Pittsburg compound B [44]. In this study, a decreasing gradient of PiB binding was found from NAWM to perilesions, T2 lesions, and black holes, reflecting the greater proportion of myelin loss along these regions. In the longitudinal part of the study, the assessment of binding dynamics at the single-voxel level allowed the generation of patient-specific maps of demyelination and remyelination and the identification of individual profiles of myelin content change over time. The authors showed that even in the presence of a similar lesion load, PwMS had extremely heterogeneous profiles of demyelination and, more importantly, remyelination, with some subjects being ‘good’ and others ‘bad’ remyelinators. In terms of the clinical meaning of this finding, while the individual indices of dynamic demyelination were not associated with disability, people with the highest remyelination showed the lowest disability [44]. These PET-based results demonstrate that remyelination profiles in MS are patient-specific and may influence disease evolution, the ability to repair, and, eventually, future clinical progression.

Concerning myelin restoration in focal lesions, a key aspect of remyelination is its role in axonal protection, as shown by biological research [22,101]. However, this effect had not been proven in vivo in MS until very recently [102]. A study combining myelin PET and diffusion-weighted MR imaging assessed the potential neuroprotective role of remyelination on subsequent neurodegeneration and found that lesion remyelination at the single-lesion level affects the microstructural integrity of surrounding tissues at follow-up in PwMS [103]. Indeed, for every extra 1% of remyelination in single WM plaques, there was an increase of almost 40% in the probability of the corresponding perilesional microstructure remaining preserved over time. Looking at the regionalization of remyelination and its effects on the brain, another PET and MRI analysis found that remyelination mostly fails in regions close to the ventricles in all PwMS, leading to speculation about the possible presence of proinflammatory soluble factors in the CSF inhibiting myelin restoration [104]. Interestingly, the more severe the failure of remyelination in periventricular lesions, the greater the atrophy developing in connected cortical regions, indicating a disconnection effect in the spreading of MS pathology in the absence of tissue repair.

A key aspect of the study of brain remyelination in MS is the evaluation of its interaction with other pathological processes of the disease, which can be more specifically assessed using PET. Indeed, understanding why remyelination succeeds or fails in some patients and in specific regions in MS requires the consideration of factors beyond the oligodendroglial lineage and myelin itself [5]. As shown in animal models, the activation state of microglial cells influences remyelination: in a pro-inflammatory state, they may inhibit oligodendrocyte differentiation, while a homeostatic or anti-inflammatory state permits repair [105]. Beyond innate immune cells, the adaptive immune system also plays a significant role in repair capabilities. A recent work using translocator protein (TSPO) PET for inflammation in MS has shown that about 60% of plaques exhibit a chronic inflammatory component, which is linked to cortical atrophy, neurodegeneration, and disability progression [106], possibly through a failure to repair. Current PET research aims to further characterize this interaction between the persistence of intralesional inflammatory cells and inhibition of remyelination. Imaging investigations have shown that, together with a gradient in remyelination capacity within the MS brain, with a global failure around the ventricles [104], there is a gradient of neuroinflammation varying with proximity to the CSF from the periventricular surface towards subcortical areas in PwMS. This suggests an influence of persisting inflammation in regions close to the CSF, possibly related to its proinflammatory composition and vascular dysfunction, on remyelination deficit in MS [107]. Inflammatory alterations of the blood–CSF barrier at the choroid plexus (ChP), which is the main CSF producer, in MS versus healthy individuals, already detectable in pre-symptomatic phases [104,108,109], could contribute to the higher remyelination failure in periventricular lesions [110]. Altogether, these findings suggest the importance of combining imaging techniques to track myelin recovery and surrogate endpoints to assess neurodegeneration to comprehensively evaluate the efficacy of putative promyelinating drugs in clinical trials for MS.

Using higher-quality tracers in terms of signal-to-noise, phase II clinical studies on remyelinating compounds in MS could benefit PET outcomes to define myelin recovery over time. In this scenario, a putative promyelinating trial could be designed with a run-in phase of 4–6 months to define, on two consecutive PET scans, individual profiles of spontaneous remyelination across PwMS, followed by two-arm randomization between the treatment and the placebo. The effect of the proposed remyelinating molecule could then be assessed by comparing a third myelin PET scan, performed after 4–6 months, to the second one, performed just before randomization. A preliminary power analysis found that 32 subjects per group would be needed to detect a 20% increase in remyelination indices, setting the following statistical parameters: α = 80% and *p* < 0.05 [6]. A trial testing the promyelinating potential of Ifenprodil is currently ongoing in PwMS (NCT06330077). Outcome measures will be the variation in P100 latency according to VEP, as well as the changes in the proportion of remyelinating voxels extracted in cortical regions from MTR acquisitions and in WM lesions from [18F] florbetaben PET acquisitions.

Moving from structural repair to functional recovery, studies using fMRI in PwMS have shown that moderate-intensity exercise increases functional connectivity in specific hubs, which counteracts the decrease in structural connectivity [111]. The favorable functional reorganization following rehabilitation, reflecting neuroplasticity, is confirmed by multiple works [112,113]. However, these findings do not directly demonstrate the existence of efficient remyelination processes taking place in the affected MS brain but could serve as proof that within the involved neuronal networks, there is still some function that can be recovered, e.g., by fostering myelin repair [114]. Further works, performed on a larger scale, are needed to better disentangle the relationship between remyelination and functional connectivity modifications in PwMS in vivo.

### 3.3. Spinal Cord

Spinal cord (SC) lesions are common at all disease stages in MS, reported in about 80 to 90% of patients [115], and microstructural damage, both inside and outside focal SC lesions, can occur very early in the disease course [116]. SC involvement is an important contributor to disability, and the occurrence of at least one SC lesion at the beginning of the disease is associated with a high risk of conversion to secondary progressive MS [117]. The number of SC lesions and the SC atrophy are associated with an increased risk of reaching higher disability scores during the disease course and are risk factors independent of patient age, baseline disability scores, and brain cortical atrophy. Therefore, the SC, as frequently damaged at all disease stages and a substantial contributor to disability, would be a putative candidate structure for assessing the neuroprotective and repair mechanisms of candidate drugs. Thus, measuring the impact of interventions to enhance recovery through spinal cord imaging tools can be of great interest.

Nonetheless, in contrast to the large number of studies on myelin dynamics in the brain, few works have focused on such processes in the spinal cord. Histological studies on remyelination capacity in spinal lesions of progressive MS have demonstrated the relevance of spinal cord pathology and failure of repair in clinical disability [8]. Incomplete spinal cord remyelination was correlated with higher disability, while this association was not observed for the brain. This reappraises the importance of spinal cord remyelination in reducing disease-related clinical progression.

In vivo techniques to study SC are far less common than brain-imaging studies for several reasons, including the SC’s small diameter, the occurrence of cardiac and respiratory motion artifacts, magnetic field inhomogeneities at this level, and the presence of flow from cerebrospinal fluid, especially in the thoracic tract [118]. All these issues have limited SC imaging applications in clinical trials, for which a highly reproducible technique and available data on the mean longitudinal change rate are needed [118]. Regarding quantitative MRI, some imaging methods have been applied to SC in MS, such as myelin water imaging [119], the magnetization transfer ratio (MTR), T1 relaxometry, and diffusion tensor imaging [120,121]. Imaging studies using MTR are helping to better characterize the in vivo remyelination potential, its reproducibility and variability [122], and its clinical implications for PwMS [123]. A recently published work using MTR in the SC of PwMS suggests a key role for microstructural repair in preventing long-term disability [124].

However, even if several technical improvements have been reached, reproducibility remains a challenge and needs to be increased before including quantitative MRI of SC in clinical trials. To note, SC lesion detection is not generally included in trials since accurate and reproducible detection and segmentation methods are still lacking. Thus, new sequences have been proposed, such as 3D phase-sensitive inversion recovery and 2-inversion-contrast magnetization-prepared rapid gradient echo sequence. They have the advantage of providing T1 mapping that gives information about the microstructural damage within lesions [24,125]. Moreover, some automatic tools under study can potentially help in SC lesion detection [126].

Some evidence suggests that EP latency changes are related to demyelination and remyelination processes [127]. Even though not specific to SC damage, using an MRI lesion frequency map showed a close relationship between the EP measurement and the SC damage in patients with MS [128]. Moreover, longitudinal changes in EP appear to be more sensitive than changes in clinical assessment [129,130] and the test–retest reliability seems quite high, especially with the use of the main cortical response, for the somatosensory EP and the cortico-muscular latency for the motor EP compared to the central conduction time [131]. All those aspects make EP suitable for SC outcome assessment in clinical trials, and one current trial (NCT04539002) assessing the role of aerobic exercise for remyelination in MS uses somatosensory EP as a primary outcome and a measure of functional myelination of the somatosensory tracts.

In conclusion, SC is frequently affected during all MS stages and contributes substantially to patient disability. SC atrophy measurement can be used as an outcome in therapeutic trials to evaluate neuroprotective treatment, but reproducibility should be improved. Quantitative imaging techniques in the SC remain a challenge, and more longitudinal studies are needed. For their high reproducibility, motor and somatosensory EPs are candidate biomarkers for remyelination and neuroprotective trials in MS and should be implemented in the near future.

## 4. Fluid-Derived Biomarkers to Track Neuroprotection and Repair

Soluble Triggering Receptor Expressed on Myeloid Cells 2 (sTREM2) is a marker of microglial activation that offers insights into ongoing immune responses and potential reparative processes mediated by microglia during MS. Preclinical studies in animal models of the disease have shown that TREM2 activation in microglia promotes phagocytosis of myelin debris [132] and increases OPC density and oligodendrocyte maturation, highlighting the role of microglia in enhancing remyelination and neuroprotection [133]. Moving to human studies, elevated sTREM2 levels in the CSF of PwMS [134] related to increased microglial activity and neuroinflammation were associated with worse outcomes [135] and normalized after highly effective treatment [136]. These seemingly opposite results highlight the complexity of microglia–oligodendrocyte interactions, which depend on multiple factors influencing microglia status.

Other potential CSF biomarkers of myelin dynamics in MS include proteins and lipids that are derived from myelin breakdown. Increased CSF levels of the myelin basic protein (MBP), for example, reflect the phenomena of myelin degradation [137] during ongoing demyelination; conversely, their decrease could indirectly measure enhanced myelin preservation and reduced tissue damage [138]. Interestingly, a long-term follow-up of MBP concentrations in the CSF after autologous hematopoietic stem cell transplantation in PwMS showed MBP normalization, indicating the resolution of harmful tissue processes and greater myelin integrity [138]. Similarly, the analysis of lipid metabolic species in the CSF of PwMS identified higher concentrations of myelin-derived lipid intermediates following demyelination [139], whose decrease could reflect reduced myelin destruction.

Micro RNAs (miR) are short, non-coding ribonucleic acids that modulate gene expression, some of which have been related to remyelination in MS. They may act by regulating microglial phagocytosis of myelin debris (e.g., miR-223), as well as oligodendrocyte maturation (i.e., miR-204, miR-219, and miR-125a-3p) and myelin deposition (e.g., miR-138, miR-145, and miR-338) [140,141]. Measuring their levels in different body fluids like the CSF and blood of PwMS could therefore inform on myelin dynamics and repair processes.

Recent advancements in fluid biomarker detection for MS include high-dimensional cytometry for immune cell profiling and SomaScan technology, which enables comprehensive proteomic analyses [142,143]. These innovations facilitate a more detailed understanding of the crosstalk between immune cells and myelin repair at the molecular level, helping in the identification of potential therapeutic targets to promote remyelination and neuroprotection.

## 5. Conclusions and Future Directions

In summary, there is a huge variety of in vivo biomarkers of myelin repair in MS, such as measurements of structural remyelination, including structural MRI and PET, as well as metrics reflecting functional recovery, like evoked potentials, fMRI, and digital markers. As this review of the currently available techniques shows, it is not possible at the moment to identify one single method outperforming the others and being, therefore, eligible to be used alone since all of them have pros and cons. Rather, complex models, in which combinations of these techniques are applied, seem to be the best and most accurate approach to tracking recovery. Although several pharmacological and non-pharmacological interventions have been tested to enhance remyelination, none have been introduced so far in standard patient care. Future endeavors should focus on combining biomarkers of remyelination to assess and validate potential therapeutic interventions to foster it. Advanced neuroimaging techniques such as 7 T or even 11.7 T MRI [144] that enable more precise demonstration of the brain microstructure would hopefully facilitate this complex task.

## Figures and Tables

**Figure 1 biomedicines-13-00357-f001:**
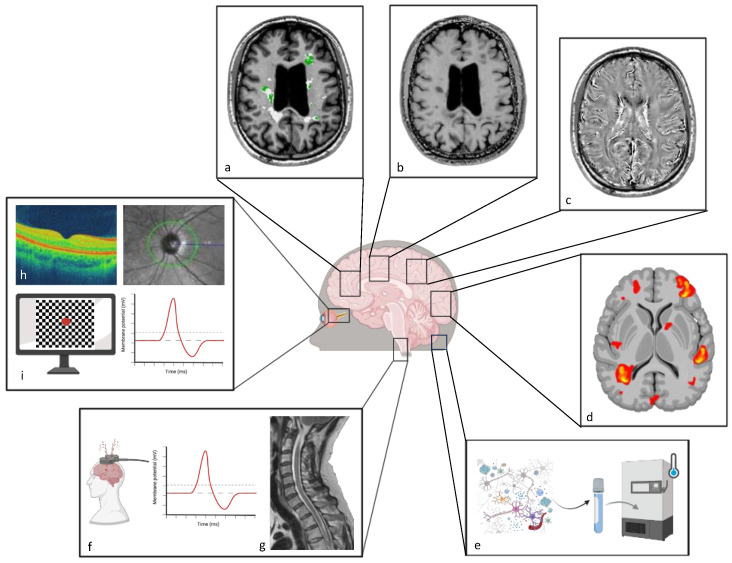
Tools to track remyelination and recovery in MS. Examples of techniques to assess myelin repair and functional recovery in vivo. (**a**) PET-derived map of remyelinated voxels (labeled in green) within MS lesions (white) overlayed on a brain 3DT1 MRI (axial plane) of a PwMS. (**b**) Brain magnetization transfer ratio image of a PwMS (axial plane). (**c**) Quantitative susceptibility mapping brain scan (axial plan) (**d**). Illustration of a functional MRI brain scan. (**e**) Schematic illustration of CSF sample analysis to measure biomarkers related to myelin dynamics. (**f**) Illustration of motor evoked potentials testing (left) and the membrane potential (right). (**g**) T2-weighted image of the spinal cord of a PwMS (sagittal plane). (**h**) Optical coherence tomography scan (left) and zoom on the optic disc at the ocular fundus assessment (right). (**i**) Visual stimuli on a video screen for visual evoked potentials recording (left) and illustration of the membrane potential (right). Created in BioRender. Camera, V. (2025) https://BioRender.com/b39y141, accessed on 29 January 2025.

**Table 1 biomedicines-13-00357-t001:** List of biomarkers of remyelination and/or functional recovery of neurons in MS.

Explored Organ/ Compartment	Tool	Biomarker
Brain	MRI	MWF, NODDI, RD, MTR, QSM, functional connectivity (matrix, distant and local connectivity density)
	PET	Voxel-based maps of myelin content change
Brain and spinal cord	MEP/SSEP	latency, amplitude
	Biosensors	Digital markers (e.g., walking speed, manual dexterity, balance)
Spinal cord	MRI	MTR
Visual system	VEP	latency, amplitude
	OCT	RNFL, GCL, IPL thickness
	Oculography	versional dysconjugacy index
CSF	ELISA, proteomic analysis, mass spectrometry, quantitative PCR	sTREM2, MBP, lipidome, miR

MRI = magnetic resonance imaging; PET = positron emission tomography; MEP = motor evoked potentials; SSEP = somatosensory evoked potentials; VEP = visual evoked potentials; ELISA = enzyme-linked immunosorbent assay; PCR = polymerase chain reaction; MWF = myelin water fraction; NODDI= Neurite orientation dispersion and density imaging; RD = radial diffusivity; MTR = magnetization transfer ratio; QSM = quantitative susceptibility mapping; RNFL = retinal nerve fiber layer; GCL = ganglion cell layer; IPL = inner plexiform layer; sTREM2 = soluble triggering receptor expressed on myeloid cells 2; MBP = myelin basic protein; miR = micro RNA.

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
