# Peer review of "Identifying Biomarkers for Remyelination and Recovery in Multiple Sclerosis: A Measure of Progress"

_biomedicines, 2025, doi:10.3390/biomedicines13020357_

Round 1

Reviewer 1 Report

Comments and Suggestions for Authors

This review explores the diagnosis, biomarkers, and treatment of multiple sclerosis, providing insights into the diagnosis and management of the disease. Some questions and suggestions are as follows:

1. What is a Non-systematic PubMed search (abstract)?

2. Are the electrical recordings in Figure 1i and Figure 1f obtained using the same technique? They look identical.

3. Unlike MWF, latency, and RNFL, fMRI does not seem to be a specific biomarker (Table 1).

4. In section 3, it is recommended to add some new discussions to broaden the readership of this article. For example, electrical stimulation can promote axonal myelination regeneration (Interdiscip. Med. 2023;1:e20230003), paracrine mechanisms of myelination regeneration (PLoS Biol. 2020, 18(12):e3000621), and EV markers (Journal of Proteomics 204 (2019) 103403).

5. There is a lack of outlook and summary of future technologies and challenges.

Author Response

We are grateful to the Reviewers for the time and effort dedicated to read and comment our paper. We have provided responses to the points raised by them, that are listed in bold. Changes in the manuscript have been indicated in track changes mode.

Reviewer 1

This review explores the diagnosis, biomarkers, and treatment of multiple sclerosis, providing insights into the diagnosis and management of the disease. Some questions and suggestions are as follows:

  1. What is a Non-systematic PubMed search (abstract)?

Thank you for this question. We apologize with the Reviewer for the lack of clarity on this aspect. We have simply conducted a narrative review based on the review of abstracts and papers identified by the search in the Pubmed database. We have now edited the corresponding part of the abstract as follows:

“Narrative review, based on a selection of PubMed articles discussing techniques to measure in vivo myelin repair and functional recovery in PwMS.”

  1. Are the electrical recordings in Figure 1i and Figure 1f obtained using the same technique? They look identical.

Both figures are describing the use of evoked potentials and they are identical indeed: the recorded signal is just a schematic illustration of a membrane potential. In the reviewed version of the manuscript, this has been clearly explicted in the legend of the figure, to take into account the Reviewer’s concern. However, Figure 1f refers to motor evoked potentials, while Figure 1i to visual evoked potentials.

  1. Unlike MWF, latency, and RNFL, fMRI does not seem to be a specific biomarker (Table 1).

Thank you for this suggestion, and we apologize for not having indicated the corresponding biomarker for fMRI. We have now removed the generic name ‘fMRI’ and replaced it with the specific fMRI metrics, being functional connectivity indices (FC matrix, distant and local fucntional connectivity density). Additionally, to make this point more clear, we have added a new sentence and reference to the part of the manuscript discussing fMRI biomarkers related to myelin content:

Regarding fMRI-derived markers of repair, some works have shown a good corre-spondence between metrics such as functional connectivity, and intracortical myelin content.” (Huntenburg JM, Bazin PL, Goulas A, et al. A Systematic Relationship Between Functional Connectivity and Intracortical Myelin in the Human Cerebral Cortex. Cereb Cortex 2017; 27: 981-997).

  1. In section 3, it is recommended to add some new discussions to broaden the readership of this article. For example, electrical stimulation can promote axonal myelination regeneration (Interdiscip. Med. 2023;1:e20230003), paracrine mechanisms of myelination regeneration (PLoS Biol. 2020, 18(12):e3000621), and EV markers (Journal of Proteomics 204 (2019) 103403).

We greatly appreciate this feedback and we agree with the Reviewer that these key ponts need to be addressed as well. We have updated section 3 to take into account the suggestions, and we have inserted the corresponding new references:

“About axonal regrowth, stimulation methods, such as non-invasive electrical stimulation (ES), are being evaluated for their potential to improve visual impairment. In the last 10 years, ES has been applied in preclinical and clinical research, thanks to the possibility of avoiding constraints imposed by drugs, such as lack of selectivity to specific lesions, al-lergic reactions or drug resistance development. Interestingly, non-invasive ES can influence neuronal polarization as well as oligodendrocyte activity, improving their differentiation. Moreover, it has been shown that increased oligodendrocyte extracellular vesicle release ameliorates axonal survival. Unfortunately, ES molecular mechanisms are only partially known and need further investigation: preclinical studies are currently being performed to discover and characterize mechanisms for axonal degeneration prevention and remyelination. Therefore, drug-based interventions currently seem to remain the quickest way to enhance recovery in the visual system. To evaluate which drug has major effects on visual system, improving remyelination and/or axonal regeneration, …”

  1. There is a lack of outlook and summary of future technologies and challenges.

We agree with the Reviewer that future perspectives should be presented at the end of the manuscript. Following his/her suggestion, we have now introduced a new final paragraph (titled “5. Conclusions and future directions”) that contains the summary and futures perspectives:

“In summary, there is a huge variety of in vivo biomarkers of myelin repair in MS, such as measurements of structural remyelination, including structural MRI and PET, as well as metrics reflecting functional recovery, like evoked potentials, fMRI and digital markers. As this review of the current available techniques shows, it is not possible at the moment to identify one single method outperforming the others and being therefore eligible to be used alone, since all of them have pros and cons. Rather, complex models, in which combinations of these techniques are applied, seem to be the best and more accurate approach to track recovery. Although several pharmacological and non-pharmacological interventions have been tested to enhance remyelination, none has been introduced so far in standard patients care. Future endeavors should focus on combining biomarkers of remyelination to assess and validate potential therapeutic interventions to foster it. Advanced neuroimaging techniques such as 7 T or even 11.7 T MRI, that enable more precise demonstration of the brain microstructure, would hopefully facilitate this complex task”.

Reviewer 2 Report

Comments and Suggestions for Authors

As there are several and growing number of drugs used in MS, as well as remielination effect is widely dissused in MS in context of drugs used there is a need to establish biomarkers of remyelination, their significance as well as in practice verify the effect of different drugs in this process.

The authors in non-systematic review performed the narrative review acording to possible biomarkers of remyelination. For better visualization of these methods I recommend to add Tables with possible biomarkers, studies as well as drugs used in MS with documented or possible effect on remyelination (1 ot 2 Tables),  

Author Response

We are grateful to the Reviewers for the time and effort dedicated to read and comment our paper. We have provided responses to the points raised by them, that are listed in bold. Changes in the manuscript have been indicated in track changes mode.

Reviewer 2

As there are several and growing number of drugs used in MS, as well as remielination effect is widely dissused in MS in context of drugs used there is a need to establish biomarkers of remyelination, their significance as well as in practice verify the effect of different drugs in this process.

The authors in non-systematic review performed the narrative review acording to possible biomarkers of remyelination. For better visualization of these methods I recommend to add Tables with possible biomarkers, studies as well as drugs used in MS with documented or possible effect on remyelination (1 ot 2 Tables)

We deeply thank the Reviewer for his/her suggestion. Since the focus of the manuscript are the different biomarkers of remyelination and recovery in MS, we agree with him/her that current methods can be more easily summarized and visualized if put in a table. This information is detailed in Table 1. Coversely, the systematic discussion of current drugs potentially promoting remyelination and the corresponding onging clinical trials for MS is not the primary scope of this paper. For information, a parallel work entirely centered on this subject is currently being prepared by the same group of Authors. For these reasons, we think that a table listing the drugs and trials to foster myelin repair and recovery would be less pertinent here, while it is already included in the other manuscript being submitted.

We hope that the Revewers will appreciate our attemps to adress their comments and suggestions.

Round 2

Reviewer 1 Report

Comments and Suggestions for Authors

Accepted.